



# Simulations of Sulfate-Nitrate-Ammonium (SNA) aerosols during the extreme haze events over Northern China in October 2014

Dan Chen[1], Zhiquan Liu[1], Jerome Fast[2], Junmei Ban[1]

[1]National Center for Atmospheric Research, Boulder, Colorado, USA

[2]Pacific Northwest National Laboratory, Richland, WA 99352

*Correspondence to:* Zhiquan Liu (liuz@ucar.edu) and Dan Chen (dchen@ucar.edu)

**Abstract:**

Extreme haze events have occurred frequently over China in recent years. Although many studies have investigated the formation mechanisms associated with $PM_{2.5}$ for heavily polluted regions in China based on observational data, adequately predicting peak $PM_{2.5}$ concentrations is still challenging for regional air quality models. In this study, we evaluate the performance of one configuration of the Weather Research and Forecasting model coupled with chemistry (WRF-Chem) and use the model to investigate the sensitivity of heterogeneous reactions on simulated peak sulfate, nitrate, and ammonium concentrations in the vicinity of Beijing during four extreme haze episodes in October 2014 over the North China Plain. The highest observed $PM_{2.5}$ concentration of 469 μg m$^{-3}$ occurred in Beijing. Comparisons with observations show that the model reproduced the temporal variability in $PM_{2.5}$ with the highest $PM_{2.5}$ values on polluted days (defined as days in which observed $PM_{2.5}$ is greater than 75 μg m$^{-3}$), but predictions of sulfate, nitrate, and ammonium were too low on days with the highest observed concentrations. Observational data indicate that the sulfur/nitric oxidation rates are strongly correlated with relative humidity during periods of peak $PM_{2.5}$; however, the model failed to reproduce the highest $PM_{2.5}$ concentrations due to missing heterogeneous reactions. As the parameterizations of those reactions is not well established yet, estimates of $SO_2$-to-$H_2SO_4$ and $NO_2$/$NO_3$-to-$HNO_3$ reaction rates that depend on relative humidity were applied which improved the simulation of sulfate, nitrate, and ammonium enhancement on polluted days in terms of both concentrations and partitioning among those species. Sensitivity simulations showed that the extremely high heterogeneous reaction rates and also higher emission rates than those reported in the emission inventory were likely important factors contributing to those peak $PM_{2.5}$ simulations.



## 1. Introduction

Anthropogenic PM$_{2.5}$ (fine particulate matter with aerodynamic diameters less than 2.5 μm) is known to play a significant role in atmospheric visibility, human health, and climate. Regional haze with extremely high PM$_{2.5}$ concentrations (exceeding tenfold of the WHO standard) has become the primary air quality concern in China, especially over the North China Plain (NCP). Severe haze pollution episodes occur frequently over the NCP, especially in Beijing (e.g. Sun et al. 2015), during almost all seasons as described by Wen et al. (2015) for the summer of 2013, Liu et al. (2013) and Yang et al. (2015) for the autumns of 2011 and 2014, and Wang et al. (2014c) and Han et al. (2015) for the winters of 2013 and 2014. Those haze events provide an opportunity to test the current understanding of sources and the formation mechanisms of PM$_{2.5}$ over the NCP as represented in current air quality models. Most air quality modeling studies have focused on winter or summer periods (e.g. Wang et al., 2014a, 2016; Wang et al., 2014d; Chen et al. 2015; Zhang et al., 2015a, Zhang et al., 2015b; Zheng et al., 2015), because emissions resulting from heat production and adverse meteorological conditions during the winter and greater photochemical production rate during the summer play crucial roles in the formation of haze in the two seasons, respectively. Although observations (Wang et al., 2014b) suggest that the biomass burning associated with autumn harvest in the NCP could be an important source of PM$_{2.5}$, but the formation mechanisms and the performance of models in autumn have not been thoroughly investigated.

Sulfate, nitrate and ammonium (denoted as SNA) are the predominant inorganic species in PM$_{2.5}$. Observations during the winter of 2013 (e.g. Wang et al., 2014c) and autumn of 2014 (Yang et al., 2015) show that SNA increases rapidly during the highest haze episodes over the NCP and makes up approximately half of the total PM$_{2.5}$ mass. Based on the studies for the winter of 2013 and the autumn of 2014, this rapid SNA increasing is associated with relative humidity (RH). There are strong correlations between RH and sulfur/nitric oxidation ratios, thus the heterogeneous reactions of precursors (sulfur dioxide SO$_2$, nitrogen oxides NO$_x$ = NO+NO$_2$) are key reactions that lead to the formation of sulfate and nitrate on the surface of particulates (e.g. Li and Shao, 2009, 2010; Li et al., 2011; Wang et al., 2012b; Zhao et al., 2013; Wang et al., 2014c; Yang et al., 2015). However, those reactions are not included in current chemical mechanisms (traditional gas-phase or aqueous-phase chemistry) in most air quality models. After identifying this deficiency, Wang et al. (2014d) and Zhang et al. (2015b) introduced and parameterized the heterogeneous uptake of SO$_2$ on deliquesced aerosols in the GEOS-Chem model and Zheng et al. (2015) comprehensively evaluated the effects of heterogeneous chemistry in the CMAQ model. Their simulations for the conditions during the 2013 winter showed great improvements when heterogeneous chemistry was included.

Precursor emissions are also important factors in determining SNA concentrations and composition. It is interesting to explore the response of SNA to precursor emission changes due to the uncertainties when applying the emission inventories in the model. Rapid changes related to historical and recent economic developments, or specific political decisions toward emission reductions are often not accounted for by emission inventories. This is particularly the case for the developing countries such as China (Richter et al., 2005). We use the latest emission inventory available for 2010, thus the changes of three precursor emissions



($SO_2$, $NO_x$, and $NH_3$) between 2010 and 2014 not be reflected in our simulations. There is amble evidence to show that $SO_2$ emissions have decreased rapidly in recent years (Lu et al., 2011; Wang et al., 2015), driven by wide implementation of flue gas desulfurization in power plants. For $NO_x$ emissions that are primarily from vehicles and industries, inventory estimates based on both bottom-up and satellite remote sensing methods show

increases before 2010 (Lamsal et al., 2011; Zhang et al., 2012; Shi et al., 2014) but different annual increase rates. Trends in $NO_x$ emissions after 2010 are not yet reported in published paper although Chinese government set the 10% $NO_x$ emission reduction target in the 12[th] Five-Year Plan (2011-2015). Among the three precursors, $NH_3$ emissions that are dominated by agricultural sources have the large uncertainty in terms of the total amount and also seasonal variations. Estimates of $NH_3$ emissions vary greatly among published papers (Streets et al.,

2003; Kim et al., 2006; Dong et al., 2010; Zhang et al., 2010; Huang et al., 2012; Xu et al., 2015). By using GEOS-Chem model, Wang et al. (2013) concluded that $NH_3$ emission plays a critical role in the SNA simulations in the NCP. In addition to the uncertainties of emission inventory, determining the dynamic variation in emissions due to unpredictable conditions (such as crop biomass burning) during the haze periods is also challenging.

In this study, we parameterized the SNA relevant heterogeneous reactions in WRF-Chem model and conducted the simulations in the NCP for a haze period in the autumn of 2014. To author's best knowledge, this is the first study that $SO_2$-$NO_2$-$NO_3$ heterogeneous reactions were taken into account in WRF/Chem model. We first evaluate the model results using the available surface observations. Then, the relative importance of precursor emissions and the missing heterogeneous chemistry on the simulated results are quantified. We also

conducted sensitivity simulations of heterogeneous reaction rates and precursor emissions. The model configuration, observations, and methodology of how heterogeneous reactions are treated are described in section 2. In section 3 and 4, the model evaluation and sensitivity analysis on heterogeneous reaction rates and precursor emissions are presented respectively. The concluding remarks are given in section 5.

## 2.  Model description, observations and methodology

### 2.1 WRF-Chem model

The Weather Research and Forecasting (WRF) model coupled with online chemistry (WRF-Chem) is based upon the non-hydrostatic WRF community model (http://www.wrf-model.org/index.php). Details of the WRF-Chem model are described by Grell et al. (2005) and Fast et al. (2006) and there have been many subsequent papers describing recent updates. We used version 3.6.1 in this study and a summary of physical

parameterization options is shown in Table 1. The model domain with a 40-km horizontal grid spacing covers most of China and the surrounding region (left panel in Fig. 1) and our interest is the NCP (right panel in Fig. 1). There are 57 vertical levels extending from the surface to 10 hPa. The WRF single-moment 6-class microphysics scheme (Grell and Devenyi, 2002) and the Grell-3D cumulus parameterization were used to treat clouds and precipitation. The Noah parameterization is used to represent land surface processes and the YSU

parameterization is used to represent boundary layer turbulent mixing (Hong et al. 2006). Initial conditions for meteorological variables are obtained from the National Center for Environmental Prediction's (NCEP) Global




Forecast System (GFS) analyses that are updated every 6 hours. The lateral boundary conditions (LBCs) for the meteorological fields are also provided by the GFS analyses. LBCs for chemistry and aerosol fields are based on prescribed idealized profiles. The simulation started on October 1, 2014 and the first three days are treated as a spin-up period and are not used in our analyses.

The Carbon-Bond Mechanism version Z (CBMZ) and Model for Simulating Aerosol Interactions and Chemistry (MOSAIC) are used as the gas-phase and aerosol chemical mechanisms, respectively, in this study. Aerosol species in MOSAIC are defined as black carbon, organic compounds, sulfate, nitrate, ammonium, sodium and chloride and other inorganic compounds. MOSAIC uses a sectional approach to represent the aerosol size distribution with 4 and 8 size bins available in the public version of the code. Research versions of

MOSAIC have used up to 20 size bins in WRF-Chem (Lupascu et al., 2015). In this study, we use 4 size bins with aerosols diameters ranging from 0.039-0.1, 0.1-1, 1-2.5, and 2.5-10 μm. The Fast-J photolysis scheme is used for photolytic rate calculations. In the standard simulation with the CBMZ-MOSIAC mechanism, the SNA aerosol formation is through oxidation and neutralization/condensation of precursor gases. The sulfate formation starts from the $SO_2$ to sulfuric acid ($H_2SO_4$) oxidation, including two pathways in the model, the gas-

phase oxidation of $SO_2$ by hydroxyl radicals (OH), hydrogen peroxide ($H_2O_2$) and ozone ($O_3$), and aqueous-phase oxidation of $SO_2$ by $H_2O_2$ and $O_3$ in clouds. The nitrate formation ($NO_x$ to nitric acid $HNO_3$ oxidation) also includes two pathways: the $NO_2$ oxidation by OH during the daytime and the newly added hydrolysis of dinitrogen pentoxide ($N_2O_5$) at night (Archer-Nicholls et al., 2014). $H_2SO_4$ and $HNO_3$ are neutralized/condensed mainly by/with $NH_3$ to form $(NH_4)_2SO_4$ and $NH_4NO_3$ respectively. As $H_2SO_4$ is nonvolatile, thus the

equilibrium surface concentration is assumed to be zero in the model. $(NH_4)_2SO_4$ is the preferential species in the completion when $H_2SO_4$ and $HNO_3$ are both present and $NH_4NO_3$ is formed only if excess $NH_3$ is available beyond the sulfate requirement. Thus the amount of $NH_3$ is a key factor in determining the SNA aerosol formation in $NH_3$-limited environment.

**Table 1.** WRF-Chem model configurations.

| | |
|---|---|
| Aerosol scheme | MOSAIC (4 bins) (Zaveri *et al.*, 2008) |
| Photolysis scheme | Fast-J (Wild *et al.*, 2000) |
| Gas phase chemistry | CBM-Z (Zavier *et al.*, 1999) |
| Cumulus parameterization | Grell 3D scheme |
| Short-wave radiation | Goddard Space Flight Center Shortwave radiation scheme (Chou and Suarez, 1994) |
| Long-wave radiation | RRTM (Mlawer *et al.*, 1997) |
| Microphysics | Single-Moment 6-class scheme (Grell and Devenyi, 2002) |
| Land-surface model | NOAH LSM (Chen and Dudhia, 2001) |
| Boundary layer scheme | YSU (Hong *et al.*, 2006) |
| Meteorology initial and boundary conditions | GFS analysis and forecast every 6 hour |
| Initial condition for chemical species | 3-day spin-up |
| Boundary conditions for chemical species | averages of mid-latitude aircraft profiles (McKeen *et al.*, 2002) |
| Dust and sea salt Emissions | GOCART |





## 2.2 Emissions

The Multi-resolution Emission Inventory for China (MEIC) (Zhang et al., 2009; Lei et al., 2011; He 2012; Li et al., 2014) for October 2010 is used as the base emission scenario. The original grid spacing of this
emission inventory is $0.25 \times 0.25$ degrees and it has been processed to match the model grid spacing (40 km). The gas precursor emissions over China for October are estimated to be 2.03 Tg for $SO_2$, 2.07 Tg for $NO_x$ ($NO_2$+NO), and 0.63 Tg for $NH_3$ in the base emission scenario. The spatial distributions of these three species are shown in Fig. 2. On the molecular basis, $NH_3$ emissions were less than the sum of $SO_2$ and $NO_x$ emissions indicating $NH_3$-limited conditions over the NCP.

The MEIC-2010 emission inventory has already been applied in some similar studies (e.g. Wang et al., 2014a, 2016; Zheng et al., 2015) for simulations over the NCP during the winter of 2013 and they found that this inventory provides reasonable estimates of total emissions from cities but is subject to uncertainties in the spatial allocations of these emissions over small spatial scales. For our simulation, uncertainties may also arise from other two aspects: the difference between the emission base year and our simulation year, and the monthly
allocations (especially for $NH_3$). To address these uncertainties, sensitivity simulations are performed that alters the emission rates as described in section 2.5.

## 2.3 Observations

The meteorological data used for all the sites in the NCP are obtained from the National Climate Data Center (NCDC) integrated surface database ([http://www.ncdc.noaa.gov/data-access/](http://www.ncdc.noaa.gov/data-access/)), shown as red circles in
Fig. 1. The following parameters were evaluated: temperature and relative humidity at 2-m (T2 and RH2), wind speed and direction at 10-m (WS10 and WD10) and 24-hr accumulated precipitation. Most of the data are of 3-hour frequency (instantaneous every 3 hours) except for precipitation. Since there is no meteorological data at sites where hourly $PM_{2.5}$ species were observed, the hourly data of T2 and RH2 from the Global Data Assimilation System (GDAS) and boundary layer height from the NOAA Air Resources Laboratory
([http://www.arl.noaa.gov/index.php](http://www.arl.noaa.gov/index.php)) at those $PM_{2.5}$ locations were used for evaluation purposes. The meteorological performance is quantified in terms of both site-by-site and also domain-wide overall statistics. The statistical measures calculated include the mean bias (MB), the root mean square error (RMSE), and the correlation (R).

The observed chemical concentrations used in the study are from three datasets: 1) the daily mean
concentration of gas phase and $PM_{2.5}$ from the Air Pollution Index (API) database in 10 cities in the NCP (shown as black dots in Fig. 1); 2) the average hourly concentrations of gas-phase pollutants and $PM_{2.5}$ at 34 monitoring sites in Beijing from the China National Environmental Monitoring Center (CNEMC); and 3) the hourly $PM_{2.5}$ measured by TEOM (tapered element oscillating microbalance, RP1405F) and 15-min species concentrations (BC, sulfate, nitrate, ammonium) in $PM_1$ measured in situ by an ACSM (aerosol chemical
speciation monitor) at the Beijing Normal University (BNU, blue dot in Fig. 1) from Yang et al. (2015). Details of the BNU instruments can be found in Sun et al. (2013). As the 34 monitoring sites in Beijing falls into 8



model grids, the observations within the same grid are averaged and then the averages of the 8 grids are compared with the model predictions. There were also 5 monitoring sites in the grid cell where the BNU site is located.  The average $SO_2$ and $NO_2$ among the 5 sites are also used for oxidation rates calculations.

## 2.4 Heterogeneous reactions

5        As discussed in section 2.1, there are several pathways for SNA formations in the standard version of WRF-Chem. Although aqueous phase processes within clouds affect sulfate, we found that the precipitation and cloud amounts in October 2014 are very low from both observation and model. Averaged 24-hr precipitation is less than 0.25 mm and 4 mm in Beijing and in NCP, respectively and the hourly cloud liquid water path are less than 10 g/m$^2$, thus the cloud-aerosol interactions were not taken into account. For the other three oxidation

pathways ($SO_2$+OH-> $H_2SO_4$, $NO_2$+OH-> $HNO_3$, $NO_2$+$N_2O_5$-> $HNO_3$), increasing the reaction constant/rates will lead to unreasonable high values during non-haze days. Thus, the original SNA formation pathways in the model are not sufficient to explain the observed sulfate concentrations.

       Following the same methodology as in Zheng et al. (2015), we added three new heterogeneous reactions into the CBMZ-MOSAIC chemical mechanism (see Table 2). These reactions are parameterized using the

pseudo-first-order rate constant and is assumed to be irreversible (Zhang and Carmichael, 1999; Jacob, 2000). The rate constant k (s$^{-1}$) for the loss of gaseous pollutants is determined by (Jacob et al., 2000; Wang et al., 2012a)

$$k_i = \left( \frac{d_p}{2D_i} + \frac{4}{v_i \gamma_i} \right)^{-1} S_p$$

Where the subscript $i$ represents the $i^{th}$ reactant for heterogeneous reactions, $d_p$ is the effective diameter of the particles (m), $D_i$ is the gas-phase molecular diffusion coefficient for reactant $i$ (m$^2$s$^{-1}$), $v_i$ is the mean molecular

speed of reactant $i$ in the gas phase, $\gamma_i$ is the uptake coefficient for reactant $i$ (dimensionless) and $S_p$ is the aerosol surface area per unit volume of air (m$^2$s$^{-3}$).

**Table 2.** Reactions and uptake coefficients added in this study

| Species | Reactions | Uptake coefficients (lower limit) | Uptake coefficients (upper limit) |
|---------|-----------|-----------------------------------|-----------------------------------|
| $SO_2$ | $SO_2$ (gas) + aerosol -> $SO_4^{2-}$ | $2.0 \times 10^{-5}$ | $5.0 \times 10^{-5}$ |
| $NO_2$ | $NO_2$ (gas) + aerosol -> $NO_3^-$ | $4.4 \times 10^{-5}$ | $2 \times 10^{-4}$ |
| $NO_3$ | $NO_3$ (gas) + aerosol -> $NO_3^-$ | 0.23 | 0.23 |

       For the most important parameter – uptake coefficients $\gamma_i$, we used a similar method as in Wang et al. (2012a) and Zheng et al. (2015). The lower and upper limits are used to present a range of $\gamma$ values in the

laboratory measurements which were applied when RH is lower than 50% and higher than 90%, respectively. The values in the 50-90% RH range are linearly interpolated based on the two limits. The lower and upper





limits of $NO_2$- and $NO_3$-related reactions are based on Wang et al. (2012a) and those of $SO_2$-related reaction are based on Zheng et al. (2015) (Table 2). We also performed sensitivity tests for those parameters as described in section 2.5.

**2.5 Scenarios**

We conducted several experiments aiming to test the model response to different heterogeneous uptake coefficients and different precursor emissions (see Table 3). The baseline emissions used in BASE and HET_BASE simulations are described in section 2.2. The heterogeneous reactions parameterization depending on RH is described in section 2.4. To determine the appropriate reaction rates, we firstly separated the impacts of $SO_2$ and $NO_2$-$NO_3$ relevant heterogeneous reactions on SNA simulations by turning off the $NO_2$-$NO_3$
relevant reactions in different simulations; and then we tested several uptake coefficients as in Wang et al. (2012a). The upper and lower values used in HET_BASE are listed in Table 2. When conducting the emission sensitivity scenarios, we took into account the trends of the national emission in recent years and their uncertainties. According to the annual National Environmental Statistical Report, the total amount of $SO_2$ emissions reached the peak in 2006 and subsequently decreased (by ~25% from 2006 to 2014 nationally). As
the MEIC-2010 emissions inventory relied on the annual statistical books in which the data is often 2-3 years older than the actual year, we assumed that the $SO_2$ emission levels in MEIC-2010 were closer to the previous 2-3 years (2007-2008). In the NCP where more strict measurements were implemented, larger-than-national-average reduction ratios were expected. For the above reasons, we applied 25% reduction of $SO_2$ in the HET_EMIS scenario. Since the NCP is in $NH_3$-limited conditions, we increased the $NH_3$ emission by 30% to
test the sensitivity in HET_EMIS.

**Table 3.** Simulation descriptions

| Simulation name | Emission | Heterogeneous reactions uptake coefficients | | |
|---|---|---|---|---|
| | | $SO_2$ | $NO_2$ | $NO_3$ |
| **For the whole month of Oct.** | | | | |
| **BASE** | BASE | Heterogeneous reactions not applied | | |
| **HET_BASE** | BASE | $2.0 \times 10^{-5}$ (lower) $5.0 \times 10^{-5}$ (upper) | $4.4 \times 10^{-5}$ (lower) $2 \times 10^{-4}$ (upper) | 0.23 |
| **HET_EMIS** | $SO_2$ decrease by 25% $NH_3$ increase by 30% | | | |
| **For Oct. 24-25** | | | | |
| **HET_MAX1** | BASE | $1.5 \times 10^{-4}$ | $6 \times 10^{-4}$ | 0.69 |
| **HET_MAX1_EMIS1** | $SO_2$ decrease by 25%; $NH_3$ increase by 50%; NO increase by 50% | $1.5 \times 10^{-4}$ | $6 \times 10^{-4}$ | 0.69 |
| **For Oct. 25** | | | | |
| **HET_MAX2_EMIS2** | $SO_2$ decrease by 25%; $NH_3$ increase by 100%; NO increase by 100% | $3.5 \times 10^{-4}$ | $6 \times 10^{-4}$ | 0.69 |





We also tested three additional scenarios for the October 24-25 period, aiming to better simulate the peak values of observed SNA aerosols during the highest polluted event. Although the emission changes in HET_EMIS may reflect some emission trends in recent years and also the uncertainty of the emission inventory, it does not dynamically represent dramatic emission increases that are possible during actual polluted events. In addition, the dependent reaction rates will be too low when the simulated RH is also too low. For these two reasons, we increased the emissions and adjusted those uptake coefficients as fixed values in HET_MAX1, HET_MAX1_EMIS1 and HET_MAX2_EMIS2.

## 3    Model evaluation

### 3.1 Meteorology

Table 4 quantifies the performance of the meteorological predictions based on the comparisons with NCDC dataset. In addition to the averaged statistics in two regions (NCP and the whole study domain), statistics for the three sites PEK, Baoding and Shijiazhuang are also listed. PEK is the only NCDC observational site in Beijing and in our simulation it is one grid cell north of the BNU site where $PM_{2.5}$ species were observed. Baoding and Shijiazhuang are two sites generally upwind of Beijing and considered as the major $PM_{2.5}$ sources in the region. The temperature and relative humidity at 2-m are overall underestimated in both the NCP (with biases of -0.09 degree for T2 and -7.09% for RH2) and the whole domain (-0.78 degree and -1.91% respectively), while wind speed at 10-meter is overestimated with biases of 0.99 m s$^{-1}$ (27.4%) over the NCP and 0.78 m s$^{-1}$ (22.6%) over China. The averaged correlations in the two regions ranged from 0.91-0.93 for the 2-m temperature (T2), 0.73-0.79 for the 2-m relative humidity (RH2), and 0.52-0.53 for the 10-m wind speed (WS10). These statistics suggest that overall meteorological performance of the model is reasonable for the NCP and the whole domain.

For site-by-site comparisons, the monthly-mean 2-m relative humidity biases at PEK, Baoding and Shijiazhuang are -11.80, -20.29 and -16.10% respectively, which are larger than the NCP average. The biases in the 2-m temperature are also larger at the three sites (from -0.24 to 1.10 degrees). The 10-m wind speed biases are relatively lower at PEK (0.36 m s$^{-1}$) and Baoding (0.37 m s$^{-1}$) than the NCP average, but higher than the average at Shijiazhuang (1.50 m s$^{-1}$).

To clearly show the fluctuations of these meteorological parameters, the time series (in local time) of T2, RH2, WS10 and WD10 at the PEK site are shown in the left panels of Fig. 3. To illustrate the relationship of meteorology and $PM_{2.5}$, the time series of T2, RH2, PBLH and $PM_{2.5}$ at the BNU site are also shown in Figure 3 (right panels). From the comparisons at both sites we can see that the T2, WS10 and WD10 simulations in Beijing are reasonable, but model failed to reproduce high RH for some days.  Large low biases of 20-30% were produced for October 13-20, 25, and 29-31.  The lowest bias occurred on October 25 when the observations RH were 70-100% and the simulated values were around 50-70%. For boundary layer height, the model successfully captured the observed temporal variations, including the relatively low boundary layer height during most of the days and also the rapid increases (along with strong winds) on October 5, 12, 15 and





26. The time series of both the observations and simulations show strong anti-correlations for boundary layer height and PM$_{2.5}$ but correlations for boundary layer height and 10-m wind speed, which indicates that boundary layer height and wind patterns are the important meteorological factors that contributed to PM$_{2.5}$ accumulation in Beijing.

**Table 4.** Statistics of meteorological simulations

| | N pairs of data | Mod. | Obs. | BIAS(Mod.-Obs.) | RMSE | R |
|---|---|---|---|---|---|---|
| **T2 (K)** | | | | | | |
| PEK | 670 | 286.26 | 286.50 | -0.24 | 1.90 | 0.90 |
| Baoding | 196 | 288.57 | 287.46 | 1.10 | 2.43 | 0.88 |
| Shijiazhuang | 196 | 288.41 | 288.79 | -0.37 | 2.80 | 0.80 |
| NCP average | 16173 | 285.28 | 285.38 | -0.09 | 2.57 | 0.91 |
| China average | 98984 | 287.03 | 287.82 | -0.78 | 3.25 | 0.93 |
| **RH2 (%)** | | | | | | |
| PEK | 670 | 54.87 | 66.67 | -11.80 | 18.48 | 0.83 |
| Baoding | 196 | 50.20 | 70.49 | -20.29 | 26.15 | 0.69 |
| Shijiazhuang | 196 | 49.72 | 65.82 | -16.10 | 23.71 | 0.64 |
| NCP average | 16170 | 54.26 | 61.35 | -7.09 | 16.44 | 0.79 |
| China average | 98968 | 62.28 | 64.18 | -1.91 | 16.21 | 0.73 |
| **WS10 (m/s)** | | | | | | |
| PEK | 638 | 2.53 | 2.17 | 0.36 | 1.39 | 0.53 |
| Baoding | 147 | 2.38 | 2.01 | 0.37 | 1.20 | 0.56 |
| Shijiazhuang | 150 | 2.74 | 1.24 | 1.50 | 2.00 | 0.49 |
| NCP average | 14781 | 3.61 | 2.62 | 0.99 | 2.10 | 0.53 |
| China average | 91243 | 3.45 | 2.67 | 0.78 | 2.09 | 0.52 |
| **WD10 (degree)** | | | | | | |
| PEK | 480 | 168.51 | 157.75 | 10.76 | 144.97 | 0.17 |
| Baoding | 147 | 162.14 | 158.16 | 3.98 | 117.40 | 0.34 |
| Shijiazhuang | 150 | 181.38 | 199.87 | -18.48 | 120.55 | 0.30 |
| NCP average | 14552 | 188.37 | 188.09 | 0.29 | 112.66 | 0.40 |
| China average | 88387 | 165.19 | 182.36 | -17.17 | 125.28 | 0.34 |
| **24-hr Precipitation (mm)** | | | | | | |
| NCP average | 71 | 2.29 | 3.52 | -1.23 | 7.35 | 0.35 |
| China average | 603 | 3.98 | 4.17 | -0.19 | 11.14 | 0.28 |

## 3.2 PM$_{2.5}$ and gas-phase pollutants

To evaluate the performance of the transport and chemistry in the original model configuration, the observed 24-hour averaged PM$_{2.5}$ concentrations at 10 national monitoring sites in the NCP are compared with the BASE simulation in Fig. 4. While Fig. 1 shows that PM$_{2.5}$ pollution has regional variations over the NCP,

10 Fig. 4 illustrates four pollutant events with peak values on October 9, 19, 24, and 31 that occurred on almost the same days at those 10 cities. The correlations are most obvious for Beijing, Baoding, and Shijiazhuang. The



BASE simulation reproduced the overall $PM_{2.5}$ levels and also the four peak events at these sites, but the peak values were too low. The daily mean $PM_{2.5}$ values at the three aforementioned cities are highest among the 10 cities and bias in the model is also the largest.

Figure 5 shows the time series of observed and simulated hourly $PM_{2.5}$, $PM_{10}$, and four trace gases in the vicinity of Beijing. The observational data are averages among 34 sites. Observed $NO_2$ and carbon monoxide (CO) show four pollution events that are consistent with the high $PM_{2.5}$ events. CO is as an important tracer since the local ambient concentrations depend mostly on emission rates, transport, and turbulent mixing. Simulated CO level is reasonable for relatively clean days when observed CO is less than 0.7 mg m$^{-3}$, but significantly underestimates peak values by 50-70% for the four polluted events. Since there boundary layer heights and winds are reasonably simulated (Fig. 3), it is possible that the CO emission rates were higher during these events which were not reflected in the baseline emission scenario. If we assume that CO emissions are underestimated in the baseline emission scenario for the peak days, we should also expect $NO_x$ emissions to be underestimated for these days since they both are emitted primarily from vehicles. However, the simulated $NO_2$ is reasonable and is too low only on October 17-18. While simulated $O_3$ is too low during the four peak events that could suggest that $NO_x$ (=NO+NO$_2$) emissions are too low, $O_3$ formation is more complex and also involves photochemistry associated with volatile organic compounds (VOC) precursor emissions. For $SO_2$, the major sources are from combustion (during heating season) and industry. Severe overestimation of $SO_2$ concentrations and underestimation of other pollutants may indicate that the $SO_2$ emissions from industry are overestimated for the October simulation in Beijing, consistent with new regulations applied in recent years. These comparisons not only serve as evaluation of the simulated chemistry, but they also provide some insights on how uncertainties in the emissions estimates affect the simulated values.

## 4 SNA response to heterogeneous reaction rates and precursor emissions

The observations at the BNU site provided not only hourly $PM_{2.5}$ concentrations, but also concentrations of black carbon, ammonium, nitrate, and sulfate, while the latter three are valuable to investigate 1) the impacts of newly added heterogeneous reactions to SNA and $PM_{2.5}$ simulation in the model and 2) the relative role of heterogeneous reactions and precursor emissions.

Figure 6 shows the hourly $PM_{2.5}$ concentrations as well as the $PM_1$ BC, sulfate, nitrate, and ammonium concentrations between October 15-31 at the BNU site. This period covers three of the highest polluted events when $PM_{2.5}$ concentrations started to increase from very clean conditions to larger than 200-400 μg m$^{-3}$. The BASE simulation reproduced the October 27-31 event in terms of $PM_{2.5}$ and the four species, but it failed to reproduce the October 16-21 and October 21-25 events. The observed BC in $PM_1$ clearly shows the increase of primary BC emissions during the two events that the model failed to reproduce, indicating a likely underestimation of the primary $PM_{2.5}$ emission during the two events. Comparisons of SNA aerosols for the October 16-21 event show that the BASE simulation captured the SNA aerosol species with small low biases on October 20. During this event, the underestimation of $PM_{2.5}$ might be due to organic compounds and other





inorganic species. While observed SNA aerosols increased dramatically starting on October 24 and almost doubled (nitrate and ammonium) or tripled (sulfate) within 48 hours during the October 21-25 event, the BASE simulations severely underestimated the SNA aerosols, especially for sulfate.

Compared with the BASE simulation, the newly added $SO_2$ heterogeneous reactions in HET_BASE simulation did increase the $PM_1$ sulfate concentrations with the largest increase occurring on October 23 when sulfate almost doubled from the BASE to HET_BASE simulations. Since the reaction rates are RH dependent, simulated sulfate was overestimated when the RH is too high on October 23. While the simulated RH is too low (Fig. 3) for the observed peak SNA event on October 24-25, the sulfate in HET_BASE did not improve too much. The sensitivity simulation results for this two-day period will be discussed later. As the competition of $SO_4^{2-}$ and $NO_3^-$ to form sulfate and nitrate respectively in $NH_3$-limited conditions, new $NO_2$-$NO_3$ heterogeneous reactions should also be added in the model along with the $SO_2$ heterogeneous reactions to avoid the nitrate decrease. Compared with the BASE simulation, nitrate in HET_BASE remained nearly the same and ammonium increased slightly. Although HET_BASE changed the SNA ratios, especially the sulfate to nitrate ratio, the total $PM_{2.5}$ mass did not increase significantly. This is because Beijing is in $NH_3$-limited condition and the SNA mass is highly dependent on the $NH_3$ emissions. In HET_EMIS, the 30% $NH_3$ emission increase leads to noticeable nitrate and ammonium increases when compared to HET_BASE. Moreover, the 25% $SO_2$ emission decrease in HET_EMIS leads to slight sulfate decrease. The total $PM_{2.5}$ mass increases in HET_EMIS when compared with BASE and HET_BASE, but HET_EMIS still has a large low bias between October 24-25. The mass fraction, depicted as a pie chart in Fig. 7, more clearly illustrates the differences in the SNA ratio changes among the simulations. In this figure, composition concentrations on polluted days (when observed $PM_{2.5}$ is larger than 75 μg m$^{-3}$) are averaged between October 15-31. In the BASE simulation, the sulfate and nitrate fractions are significantly lower on polluted days. With the newly added heterogeneous reactions, the SNA fractions from HET_BASE are very close to observations.

The sulfur oxidation ratio (SOR= $nSO_4^{2-}/(nSO_4^{2-}+nSO_2)$) ($n$ refers to the molar concentration) and the nitric oxidation ratio (NOR= $nNO_3^-/(nNO_3^-+nNO_2)$) are important factors showing the gaseous species oxidation rates and the secondary transformation (Sun et al., 2006, 2013). The high fractions of sulfate and nitrate in heavily polluted episodes could be related to the high oxidation rates of SOR and NOR. Figure 8 shows the observed and simulated SOR and NOR for October 15-31 at the BNU site. The colors of those scatter points are associated with $PM_{2.5}$ concentrations. As simulated $SO_2$ were 2-3 times higher in Beijing compared with observations from the 34 local sites (Fig. 3) possibly due to an overestimates in the $SO_2$ emissions, the calculated SOR by the simulations would be artificially low due to the $SO_2$ overestimation. To correct this problem and to check the SOR differences among different simulations, we used the observed $SO_2$ when calculating the SOR for the BASE, HET_BASE and HET_EMIS simulations. The hypothesis is that the environment in Beijing is sulfur rich and the $SO_2$ overestimation would be corrected by reducing the $SO_2$ emissions in the model. From the observations we can see that SOR and NOR have a drastic increase in the 80-100% RH range and are strongly correlated with high $PM_{2.5}$ concentrations. In the BASE simulation without the heterogeneous reactions, the corrected SOR and NOR are within a reasonable range when RH is below 60%,



but the corrected SOR and NOR are too low when RH is between 80-100% and 90-100%, respectively. SOR and NOR are improved in HET_BASE simulation. In HET_EMIS simulation, the increase of $NH_3$ emissions and decrease of $SO_2$ emissions lead to increased NOR and $PM_{2.5}$ total mass.

From the October 15-31 period (Fig. 6-7), we can see that the BASE simulation generally reproduced the $PM_{2.5}$ mass and also SNA species in relatively clean days, and the added heterogeneous reactions in HET_BASE helps to improve the SNA simulations (especially for sulfate) during the polluted events. Nevertheless, there are still large biases between October 24 and 25. Given that the simulation meteorology is reasonable for these two days (Fig. 3), we assume that there might be two possible reasons for the biases: 1) the reaction rates are still too low due to the settings in the upper limit of $SO_2$ uptake coefficients and also the simulated RH, 2) the increase of precursor emissions in the upwind areas (southwestern of Beijing) are not reflected in the model as small fires from autumn biomass burning are not updated in the emission inventory while fire count from http://rapidfire.sci.gsfc.nasa.gov/cgi-bin/imagery/firemaps.cgi did show intensive fire locations in southern Heibei, which is upwind of Beijing. Another factor is that the underestimations of $PM_{2.5}$ in Shijiazhuang and Baoding on October 24-25 are even larger. To examine our assumptions, several different scenarios are (details in section 2.5) simulated for these two days and the results are shown in Fig. 9. In HET_MAX1 simulation, the reaction rates for the $SO_2$-$NO_2$-$NO_3$ heterogeneous reactions are fixed and tripled of the upper limits in HET_BASE. Without any emission increase, simulated sulfate almost reached the peak values on October 24 but were still underestimated on October 25. A higher $SO_2$ heterogeneous reaction rate (7 times of the upper limit in HET_BASE) and also doubled $NH_3$ emissions in HET_MAX2_EMIS2 enable the model to reach the sulfate peak on October 25. For nitrate, the increase of $SO_2$ heterogeneous reaction rates in HET_MAX1 lead to lower nitrate concentrations even though the $NO_2$-$NO_3$ heterogeneous reaction rates were also tripled. Only when $NO_x$ and $NH_3$ emissions in HET_MAX1_EMIS1 are increased by 50% do simulated peak nitrate concentrations become comparable with observations on October 24 and 25. Therefore, to reach the peak values of SNA aerosols on these two days, the sensitivity simulations suggest an increase of $SO_2$ heterogeneous reaction rates and $NO_x$ and $NH_3$ emissions are essential. For the best simulations (HET_MAX1_EMIS1 on October 24 and HET_MAX2_EMIS2 on October 25), total $PM_{2.5}$ mass was improved by ~100 μg m$^{-3}$ compared to the BASE simulation. We note that transport of SNA aerosols from upwind areas might be another important contributor in addition to precursor emission rates.

## 5   Conclusions

Accurately predicting the concentration and composition of particulate matter is still very challenging for climate and air quality models. In this study, the WRF-Chem model was used to simulate the high $PM_{2.5}$ events in the North China Plain (NCP) surrounding Beijing for an autumn period (October 1-31, 2014). Our objective is to evaluate the capability of one such model, better understand the mechanisms that form sulfate, investigate the uncertainties associated with a set of heterogeneous chemical reactions, and improve the simulations of very high $PM_{2.5}$ concentrations during pollution episodes. The evaluations of meteorological parameters in the NCP show that model is capable to capture the temporal variations of boundary layer height



as well as the low boundary layer heights during four pollution events. The deficiencies in meteorological forecasts were underestimates in relative humidity, especially during the most polluted days, and the overestimates in wind speed. While the default version of the CBMZ-MOSAIC mechanism available in the public version of WRF-Chem was able to simulate the high $PM_{2.5}$ concentrations (daily mean up to 200 μg m$^{-3}$)

for most of the cities in the NCP, the model severely underestimated the peak values (hourly mean greater than 400 μg m$^{-3}$) in Beijing and at upwind sites (Baoding and Shijiazhuang). PM species observations at BNU site in Beijing show that the sulfate-nitrate-ammonium aerosols (SNA) increases dramatically during these peak events and the increased sulfur-oxidation rates and nitric-oxidation rates are strongly correlated with the high relative humidity (80-90%) on those days.

The failure of the model to simulate the peak $PM_{2.5}$ concentrations is mainly due to the underestimation of SNA and secondary organic compounds. Analyses of the SNA underestimation revealed that missing $SO_2$-$NO_2$-$NO_3$ relevant heterogeneous reactions in the current aerosol scheme are likely important in China. Following the methodology in Zheng et al (2014), the RH-dependent $SO_2$, $NO_2$ and $NO_3$ uptake heterogeneous reactions were added to the CBMZ-MOSAIC chemistry scheme. With the newly added reactions, the SNA

simulations of the ratios of SNA in PM and the partitioning of sulfate and nitrate aerosols were improved on polluted days. However, there was still a 100 μg m$^{-3}$ underestimation of SNA aerosols for the October 24-25 period when $PM_{2.5}$ concentrations were as high as 400-500 μg m$^{-3}$. Two possible explanations are proposed: 1) The RH-dependent reaction rates of those heterogeneous reactions especially the $SO_2$ uptake reactions on those peak days were not high enough either due to the underprediction in RH or the setting of the upper limits of

uptake coefficients; 2) Comparisons of modeled gas-phase precursors showed the possibility of precursor underestimation (especially $NO_x$) in the model. Although the two explanations cannot be proved definitively without additional observational evidence, sensitive simulations with increased heterogeneous reaction rates and increased $NO_x$ and $NH_3$ emissions show great improvement of SNA simulations in the model for the October 24-25 peak pollution events.

We conclude that RH in the 80-100% range is a significant factor contributing to peak $PM_{2.5}$ values, especially during the heavily polluted days when sulfur and nitric oxidation rates almost doubled or tripled indicating the rapid heterogeneous reactions. With the underprediction in RH in this range and the corresponding low reaction rates, it is difficult for the model to reproduce the high concentrations of SNA. Data assimilation of meteorological variables, particularly humidity, might be useful from this point of view. Two

other concerns should be addressed in future studies. First, the heterogeneous reaction rates applied in this study were based on other literatures and the species evaluation was based on only one site in Beijing. Although the comparisons at one site are improved for the polluted days in Beijing, the sensitivity simulations on peak days indicated the current setting of the upper limit reaction rates might be still too low. More species evaluations for longer periods of time are needed to determine appropriate rates. Second, precursor emissions that are higher

than available in the emission inventory due to special conditions might also be another factor contributing to the peak $PM_{2.5}$ events. Data assimilation techniques that uses the observations to estimate the dynamic emission changes might provide significant improvements in future studies.



## Acknowledgement

This work was partially funded by IBM Research China. NCAR is sponsored by the National Science Foundation. J. Fast is supported by the U.S. Department of Energy's Atmospheric System Research (ASR) program (KP17010000/57131).

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





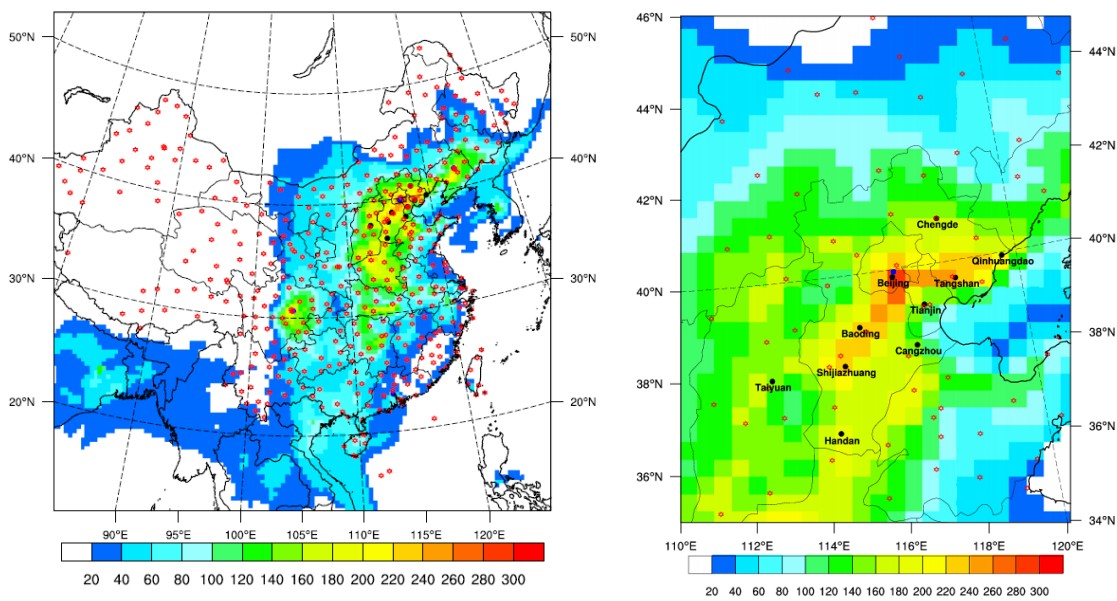

**Figure 1.** The model domain and (left panel) and the Northern China Plain (NCP, right panel). Dots are the observational sites, where red denotes NCDC meteorological sites, blue denotes the Beijing Normal University (BNU) site, black denotes AQI national monitoring sites. Shaded backgrounds are model-simulated $PM_{2.5}$ concentrations ($\mu g\ m^{-3}$) that illustrate the regional pollution on October 10.





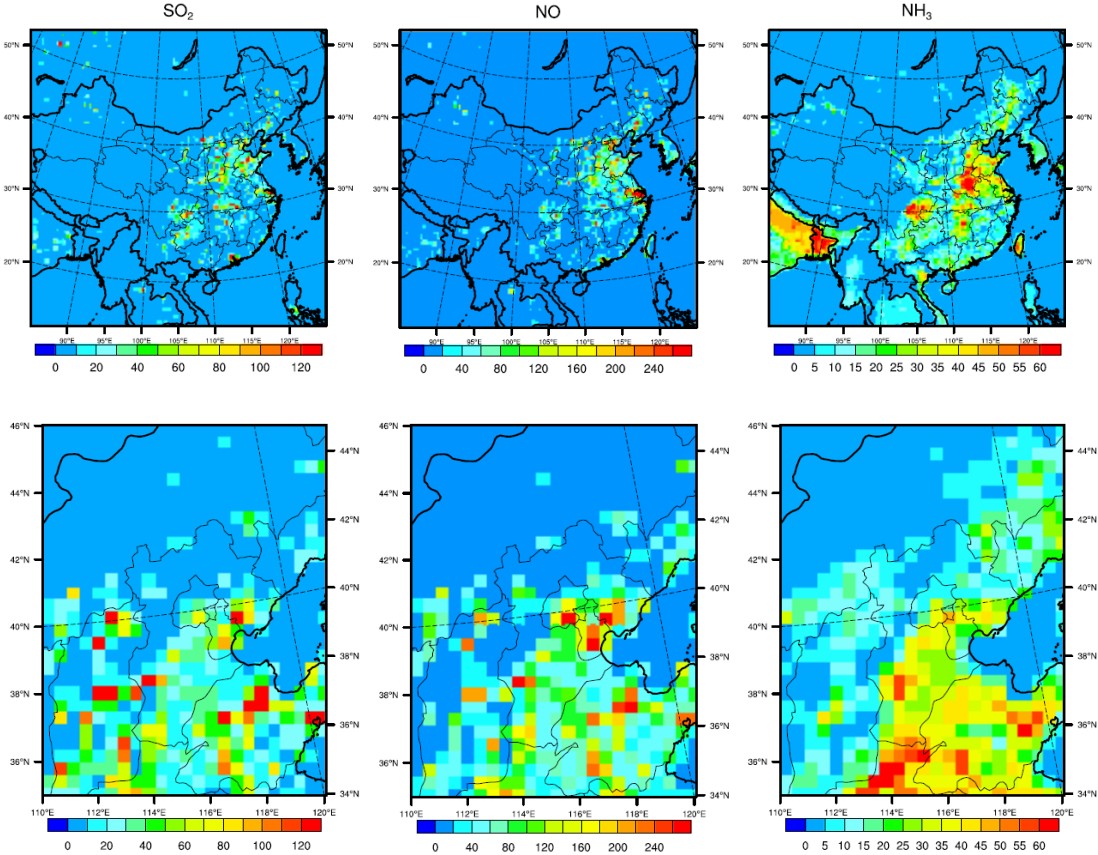

**Figure 2.** Spatial distribution of $SO_2$ (left), NO (middle) and $NH_3$ (right) emissions over the model domain (top panels) and in NCP (bottom panels). Units are in mol $km^{-2}$ $hr^{-1}$.





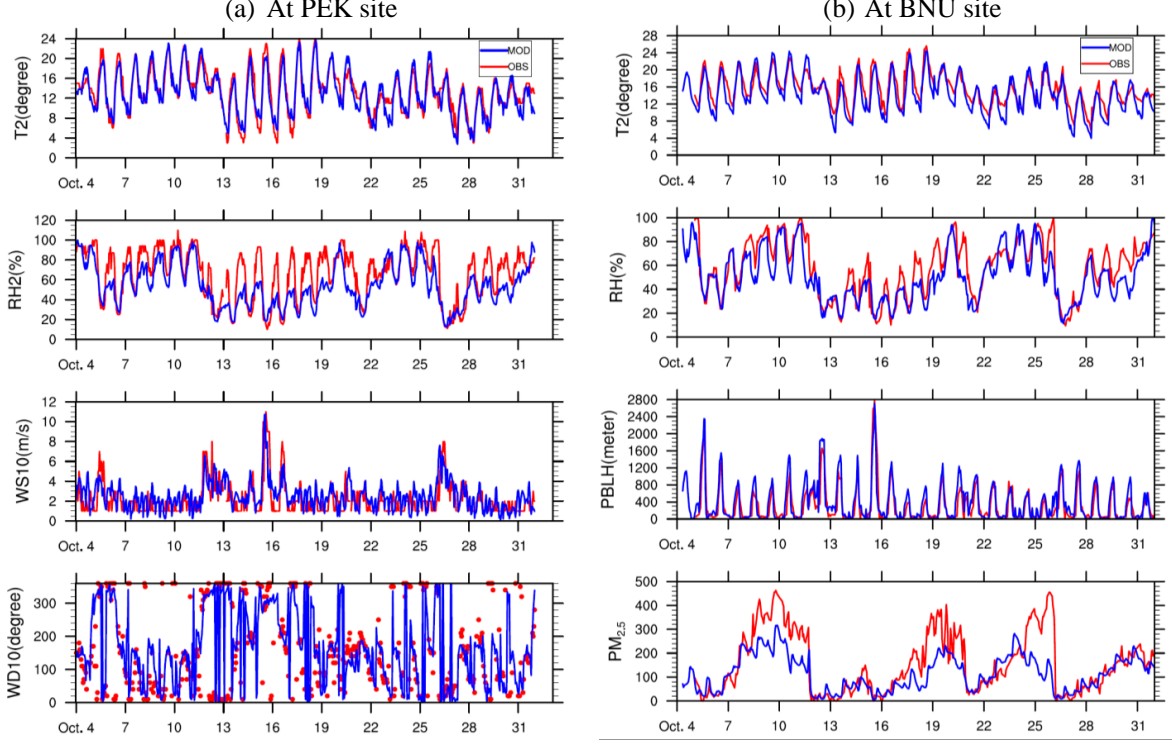

**Figure 3.** Time series of observed and simulated meteorological parameters and PM$_{2.5}$ at the (a) PEK and (b) the BNU sites (in local time). Time starts at 00:00 local time.





**Figure 4.** Time series of observed 24-hour averaged PM$_{2.5}$ concentrations (μg m$^{-3}$) at 10 national monitoring system sites in the NCP (locations as black dots in Fig. 1), along with the results from the BASE simulation.



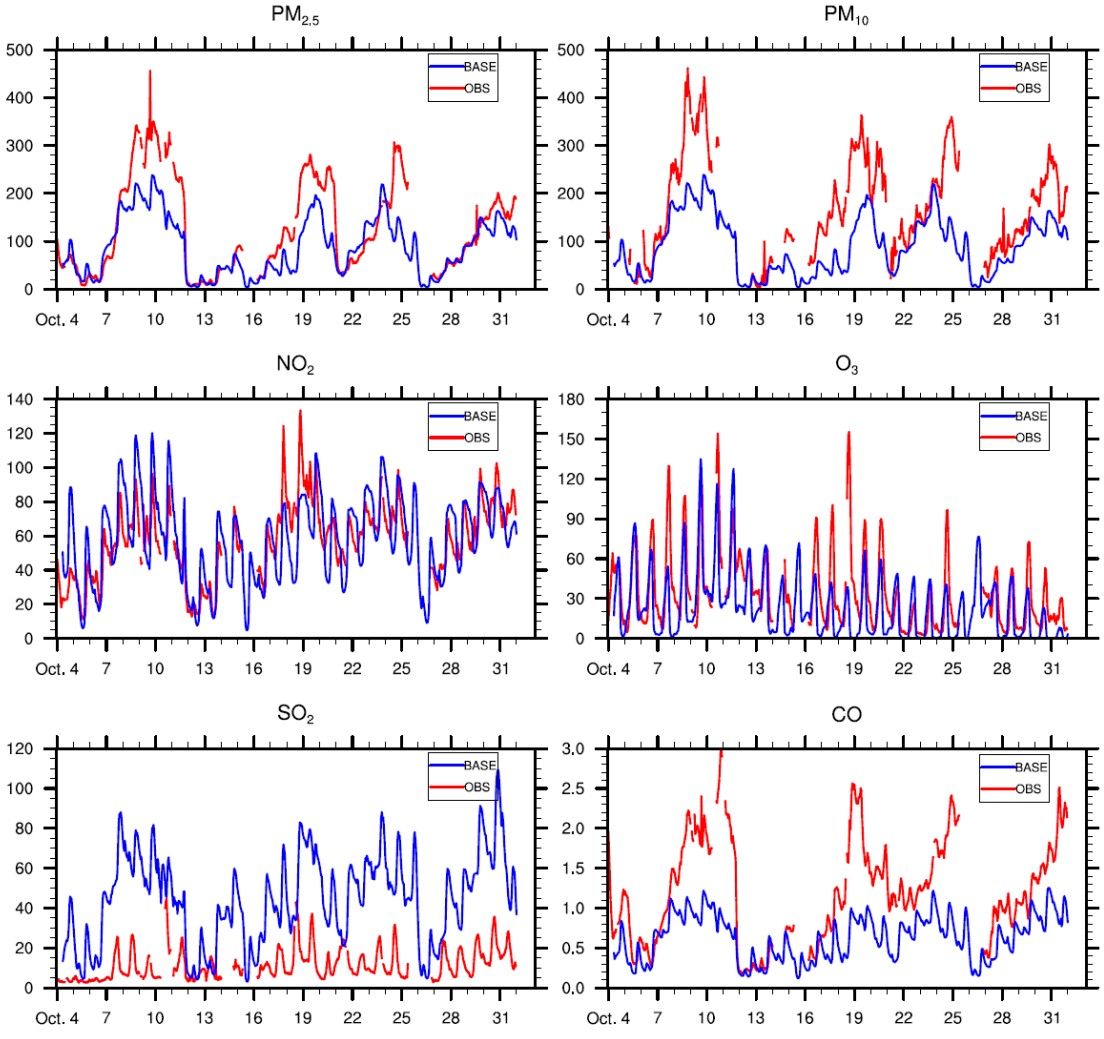

**Figure 5.** Time series of observed averaged pollutants concentrations at 34 sites in Beijing (from local monitoring system) along with the results from the BASE simulation. Units are mg m$^{-3}$ for CO and μg m$^{-3}$ for the other panels.



**Figure 6.** Time series of observed (OBS) and simulated (BASE, HET_BASE, HET_EMIS) hourly PM$_{2.5}$ and PM$_1$ species ( µg m$^{-3}$) including black carbon (BC), sulfate (SO$_4$), nitrate (NO$_3$) and ammonium (NH$_4$). Descriptions of model scenarios are given in Table 3.





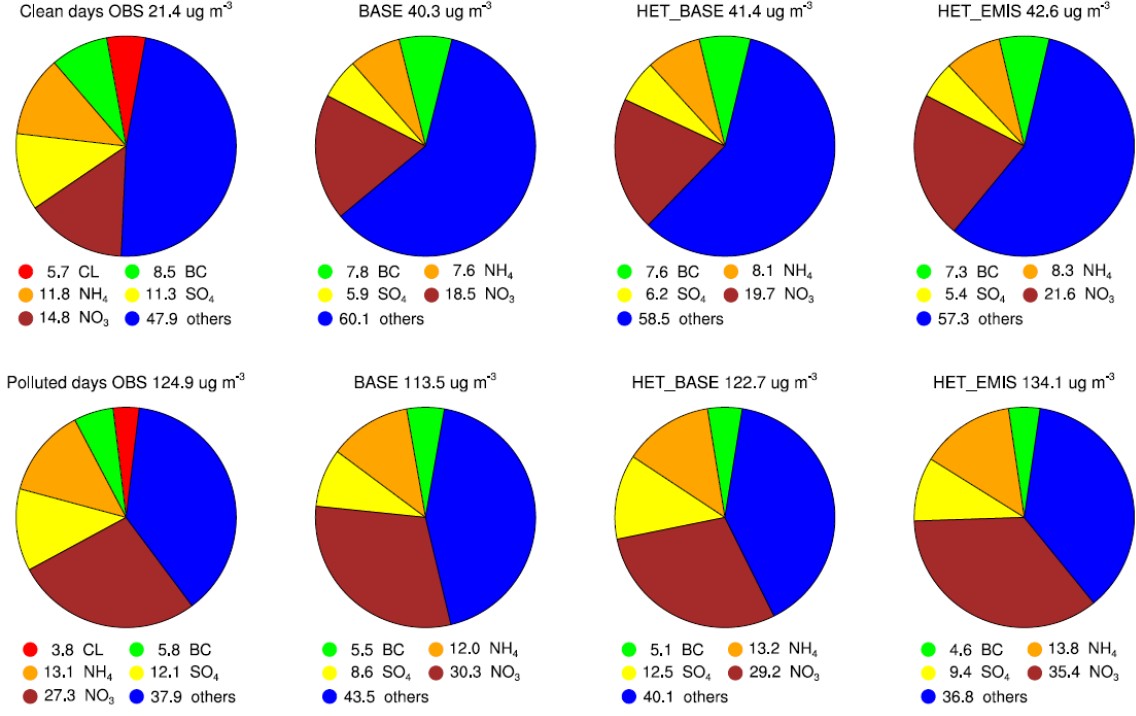

**Figure 7.** Pie charts of observed and simulated mass fractions of $PM_1$ species on relatively clean days (observed $PM_{2.5} < 75$ μg m$^{-3}$) and polluted days (observed $PM_{2.5} >= 75$ μg m$^{-3}$) between the October 15-31. The units of total $PM_1$ concentrations (listed in panel titles) are μg m$^{-3}$. The species fractions (listed in panel labels) are in percentage (%).





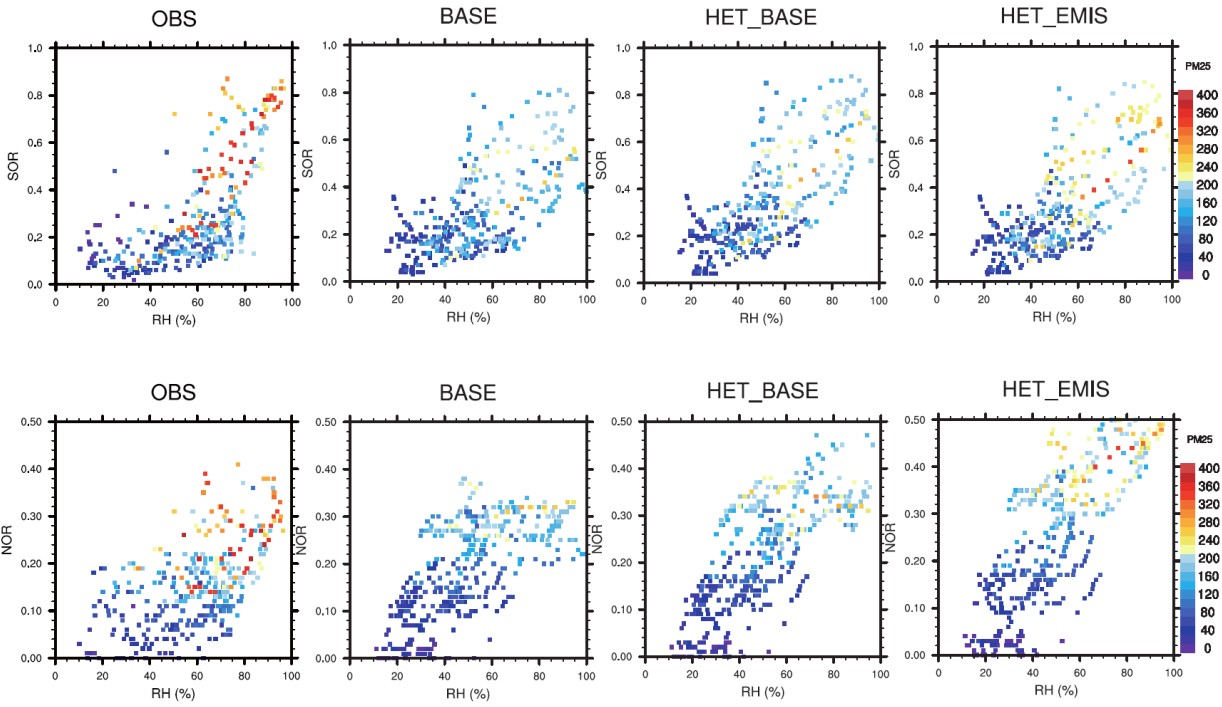

**Figure 8.** Observed (OBS) and simulated (BASE, HET_BASE, HET_EMIS) sulfur (SOR) and nitric (NOR) oxidation rates in $PM_1$ between October 15-31. The x-axis is the observed for left panels and simulated relative humidity for the other panels. Colors denote different $PM_{2.5}$ concentrations ($\mu g/m^3$).





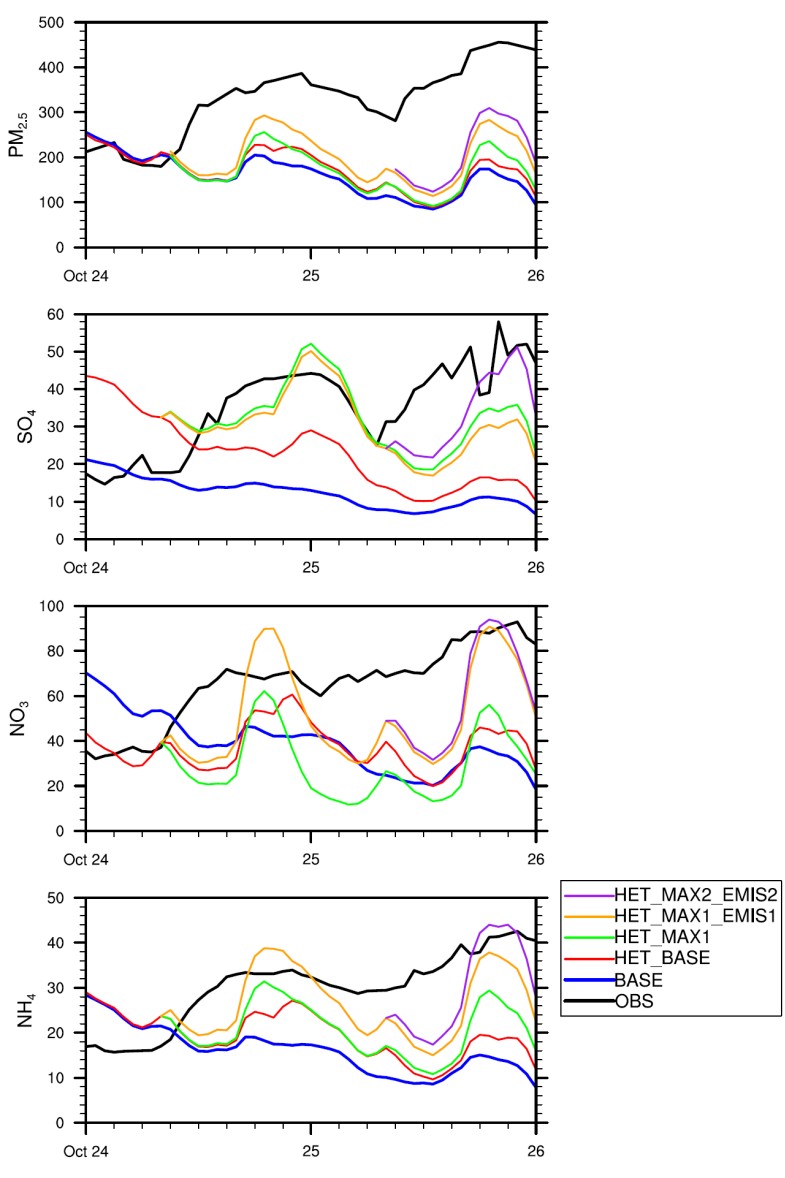

**Figure 9.** The sensitivity simulations of $PM_{2.5}$ and SNA species ($\mu g\ m^{-3}$) for the October 24-25 pollution period.