# Peer review of "Simulations of Sulfate-Nitrate-Ammonium (SNA) aerosols during the extreme haze events over Northern China in October 2014"

_Atmospheric Chemistry and Physics, 2016_

## Referee Comment (RC1) · Anonymous Referee #2 · 4 Jun 2016

The manuscript discussed the formation of inorganic aerosols (sulfate, nitrate and ammonium) over the North China Plain (NCP) in October 2014 when several extreme haze events occurred. The authors used the WRF-Chem meteorology-chemistry model to interpret surface measurements of meteorology, air pollutants, and aerosol composition during the period. A suite of sensitivity simulations was conducted to quantify the impacts of heterogeneous reaction rates and precursor emissions to inorganic aerosols. The results show that for the haze events in October 2014 over the NCP high heterogeneous reaction rates and high precursor emissions under high relative humidity are likely important factors for the peak PM2.5 concentrations.

This study fits the scope of ACP by targeting the chemical mechanism of inorganic

aerosol formation in a pollution hotspot. The manuscript is clearly presented. I have several comments below that I think the authors shall address before considering publish.

**Specific Comments:**

1) Page 4, WRF-Chem description: Do you consider aerosol-meteorology interactions in the model simulations? You mentioned that cloud-aerosol interactions were not taken into account in section 2.4. How about radiative effects? Please clarify.

2) Page 5, Line 7: Please add the emission totals over the NCP. This would help to understand the statement in the next sentence that on the molecular basis NCP is NH3-limited.

3) Page 7, Line 8-10: I do not see where in the text you discussed the simulation with only the SO2 heterogeneous reaction. The simulation is also not listed in Table 3. Please clarify.

4) Page 7, Line 17-20: Please clarify whether the emission perturbations (e.g., 25% decrease in SO2, and 30% increase in NH3) are applied to the whole modeling domain or just over the NCP. After decreasing SO2 emissions and increasing NH3 emissions, is the NCP area still under NH3-limited condition?

5) Page 8, Line 5-7: I suggest add some sentences explaining how you determine those high uptake coefficients, for example, to increase SO2 uptake coefficient by a factor of 10. Would those values be valid in the real atmosphere?

6) Page 8, section 3.1: It appears to me that this section is missing some discussions on how biases in simulated meteorology would impact the aerosol simulation. The model generally underestimates relative humidity, while overestimates surface wind speed. How would it affect the aerosol simulation?

7) Page 10, Line 17: Please quantify how much percentage SO2 is overestimated in the model. Can the model versus measurements differences be explained by the

recent emission trends? Please clarify.

8) Page 11, second paragraph: Some discussions on the use of observed SO2 con-
centrations to calculate model SOR are needed. Despite the sulfur rich environment,
reducing SO2 emissions in the model not only reduces SO2 concentrations, but also
aerosol sulfate concentrations. How would SOR respond to SO2 emission changes in
the model? This can be evaluated with the simulation with 25% SO2 emission reduc-
tion. I suggest add some sentences discussing the uncertainties in the model SOR
values.

9) Page 25, Line 25 "We conclude that RH in the 80-100% range is a significant factor
contributing to peak PM2.5 values". The conclusion is only partly true. In the 80-100%
range SOR and NOR values are much higher, but as for the peak PM2.5 values, from
Figure 8, it appears that there are comparable amounts of high PM2.5 values in the
60-80% range. Please clarify.

**Technical Comments:**
1) Page 9, Line 2-4 "But correlations for boundary layer height and 10-m wind speed",
missing some words here? What correlations?

2) Page 10, Line 9 "Since there" should be "Since their"?

3) Page 18, Figure 1 The blue symbol and the city labels are too small to read on the
Figure. Please make them larger.

4) Page 20, Figure 3 Please describe in the Figure caption what are those meteorolog-
ical variables, such as T2, RH2, WS10, and WD10.

---

## Referee Comment (RC2) · Anonymous Referee #1 · 11 Jun 2016

The manuscript by Chen et al. considers factors that may help explain deficiencies in WRF-Chem simulations of secondary inorganic aerosol in extreme haze events in China. Following studies with other models, they consider additional heterogenous reactions, and perform sensitivity studies to evaluate the impacts of these reactions as well as uncertainties in emissions. The topic is timely and of importance / relevance for ACP. The paper is generally well written, despite some grammatical issues. More quantitative comparison could be made to recent papers that have evaluated SO2 and NO2 trends in this region, or that have estimated the contribution of different aerosol precursor emissions to PM2.5 in Beijing. The final model performance is indeed much better, although there is still room for improvement in the model and in our understanding of these haze events. I recommend publication following revisions to address the comments below.

Comments:

General: This work seems to still be missing a key reaction, which is aqueous-phase oxidation of S(IV) (the sum of dissolved SO2, HSO3−, and SO32−) by dissolved nitrogen dioxide (NO2) that has been documented in the literature (Lee and Schwartz 1983, Clifton et al 1988, Sarwar et al 2013). As shown in Zhang 2015b, this made a substantial improvement to GEOS-Chem (in ways which would likely similarly improve the WRF-Chem) beyond the heterogenous reactions that are considered here. Thus, I would also suggest the authors include this reaction in their analysis as an additional sensitivity calculations.

3.5: There are several recent papers on SO2 and NO2 trends, for example Krotkov et al., ACP, 2015, see Fig 8, or Cui et al., ACP, 2016. The former would be useful to compare to when considering the SO2 and NO2 emissions trends projected in this paper.

3.11: GEOS-Chem was also used to specifically quantified the role of NH3 in Zhang et al. 2015b.

3.6: Not clear what is meant by "published paper". Perhaps official report? Or bottom up inventory?

p4/Table 1: What scheme is used for calculating gas-aerosol partitioning of HNO3/NO3 and NH3/NH4?

5.8: One would reach the same conclusion in this particular case, but more rigorously the moles of NH3 should be compared to the moles of 2 x SO2 + NOx.

5.13: Recent Nature Geo paper (McLinden et al., 2016) highlights missing SO2 sources in this (or similar) inventory.

8.20: Could some comparison to other studies / domains / models be referenced here, in terms of substantiating what it to be considered a "reasonable" accuracy for this type of model? At present, that word is used rather loosely.

10.15: Well, that would depend on the NOx/VOC regime, which the authors could easily check from their modeling results.

10.17: This could also instead indicate that SO2 oxidation is too weak / slow in the model.

Section 4: It wasn't clear to me why the detailed speciated analysis was limited to only a few days. Why was this not performed for the entire month? Were the observations just not available? The peak PM2.5 concentrations earlier in the month, Oct 7 - 10, were the largest of the month, and at a time when RH was well simulated in the model. Seems like this would be a good target to include in the analysis.

11.24: The average magnitude is improved, the the temporal correlation is not likely improved. Can the authors provide a table, or perhaps just write directly on these plots, what the statistics such as $R^2$ and NMB are for these results?

Fig 7: In terms of comparing the observed to modeled % contributions from sulfate/ammonium/nitrate, it would easier to evaluate visually if the plots were of just these 3 species. At the very least, they could remove CL from the obs, so make a more direct comparison.

12.27: This could be understood more quantitatively by considering results from Zhang 2015b.

General: Did the authors try increasing the RH as a sensitivity test?

Corrections: General: I didn't type up all of the grammatical corrections; please have Jerome do a final proof-read of the article prior to resubmission.

abstract, last line: situations —> concentrations 2.4: exceeding the WHO standard

tenfold 2.15: PM2.5, the formation 3.1: 2014 may not be reflected (or are not reflected) 3.15: WRF-Chem and 3.16: conducted simulations . . . To our best knowledge 3.17: WRF/Chem model. —> WRF-Chem. 3.18: using available 3.22: analysis for 3.28: missing comma 3.35: et al., 4.19: nonvolatile, the 5.14: from two other aspects 5.36: fall into 6.8: respectively, 6.17: include equation number, comma goes directly after the equation on the same line, and then "Where" is not capitalized. 6.21: units of surface area per unit volume of air seem incorrect. 7.8: we first 7.10: simulations; we then tested. . .
* * *

---

## Author Comment (AC1) · 21 Jul 2016

**Reviewer 2#**
The manuscript discussed the formation of inorganic aerosols (sulfate, nitrate and ammonium) over the North China Plain (NCP) in October 2014 when several extreme haze events occurred. The authors used the WRF-Chem meteorology-chemistry model to interpret surface measurements of meteorology, air pollutants, and aerosol composi- tion during the period. A suite of sensitivity simulations was conducted to quantify the impacts of heterogeneous reaction rates and precursor emissions to inorganic aerosols. The results show that for the haze events in October 2014 over the NCP high heterogeneous reaction rates and high precursor emissions under high relative humidity are likely important factors for the peak $PM_{2.5}$ concentrations. This study fits the scope of ACP by targeting the chemical mechanism of inorganic aerosol formation in a pollution hotspot. The manuscript is clearly presented. I have several comments below that I think the authors shall address before considering publish.

We thank Referee # 2 for their comments and suggestions that have helped to improve this manuscript. Our responses to comments and the corresponding changes to the manuscript are detailed below in blue text. Revised manuscript is after the response letter.

Specific Comments:
1) Page 4, WRF-Chem description: Do you consider aerosol-meteorology interactions in the model simulations? You mentioned that cloud-aerosol interactions were not taken into account in section 2.4. How about radiative effects? Please clarify.
- The aer_ra_feedback option was turned on so that aerosol radiative feedback was taken into account. In section 2.4, we emphasized that the cloud-borne aerosols were not taken into account, as the cloud and precipitation amounts were negligible for that period. We have clarified this in the text.

2) Page 5, Line 7: Please add the emission totals over the NCP. This would help to understand the statement in the next sentence that on the molecular basis NCP is $NH_3$-limited.
- We have added the emission totals over the NCP. "….the values over the NCP are 0.60 Tg (9.32 Gmol) $SO_2$, 0.63 Tg (13.8 Gmol) $NO_x$ and 0.13 Tg (7.8 Gmol) $NH_3$ respectively". On the molecular basis, $NH_3$ emissions were much less than the sum of $2*SO_2$ and $NO_x$ emissions indicating $NH_3$-limited conditions over the NCP.

3) Page 7, Line 8-10: I do not see where in the text you discussed the simulation with only the $SO_2$ heterogeneous reaction. The simulation is also not listed in Table 3. Please clarify.
- Yes, we did start from the simulations with only the $SO_2$ heterogeneous reactions and we tried two sets of reaction coefficients: 1) $SO_2$_only_1 with lower/upper limits $2 \times 10^{-5}/5 \times 10^{-5}$ which are the same as in HET-BASE, and 2) $SO_2$_only_2 with lower/upper limits $1.0 \times 10^{-4}/2.6 \times 10^{-4}$. But we found that with only this $SO_2$

relevant reaction, simulated nitrate decrease due the competition of $SO_4^{2-}$ and $NO_3^-$ to form sulfate and nitrate, respectively, in NH$_3$-limited conditions (see figure below). Thus we decided to include both SO$_2$ and NO$_2$-NO$_3$ heterogeneous reactions in the HET-BASE scenario.

[Figure]

Figure S1. Same as Figure 6 but for the BASE, SO$_2$_only_1, and SO$_2$_only_2 simulations (units: µg m$^{-3}$). SO$_2$_only_1 with lower/upper limits $2\times10^{-5}/5\times10^{-5}$; SO$_2$_only_2 with lower/upper limits $1.0\times10^{-4}/2.6\times10^{-4}$.

4) Page 7, Line 17-20: Please clarify whether the emission perturbations (e.g., 25% decrease in SO$_2$, and 30% increase in NH$_3$) are applied to the whole modeling domain or just over the NCP. After decreasing SO$_2$ emissions and increasing NH$_3$ emissions, is the NCP area still under NH$_3$-limited condition?
- The emission changes were applied to the whole modeling domain. The NCP is still under NH$_3$-sensitive condition after applying the SO$_2$ and NH$_3$ emission changes (please refer to the NCP emission totals in section 2.2).

5) Page 8, Line 5-7: I suggest add some sentences explaining how you determine those high uptake coefficients, for example, to increase SO$_2$ uptake coefficient by a factor of 10. Would those values be valid in the real atmosphere?
- Actually we did not have any solid evidence to determine the SO$_2$ uptake coefficient in those sensitivity simulations due to the limited observations (only one site for a few days). What we can do is to try different values to best match the observations. In the three scenarios for Oct. 24-25, the SO$_2$ uptake coefficients were 3 and 7 times of the values in HET-BASE. However, more observations are needed to test and optimize those parameters. (Page 8, lines 8-11)
- In addition, it is still difficult to say whether the sulfate underestimation is only due to the missing of heterogeneous reactions. Zhang et al (2015b) also emphasized that aqueous reactions of SO$_2$ by H$_2$O$_2$, O$_3$ and NO$_2$ in cloud and on deliquescent aerosols would also help to improve the sulfate simulations. We have added the discussion in the text. (Page 13, third paragraph)

6) Page 8, section 3.1: It appears to me that this section is missing some discussions on how biases in simulated meteorology would impact the aerosol simulation. The model generally underestimates relative humidity, while overestimates surface wind speed. How would it affect the aerosol simulation?

- Thanks! We have added the discussion on the evaluation of the meteorological performance and also emphasized that that the overestimation of wind speed and underestimated relative humidity may lead to a negative bias of chemical species in the simulation. (Page 9, lines 2-13)

7) Page 10, Line 17: Please quantify how much percentage SO$_2$ is overestimated in the model. Can the model versus measurements differences be explained by the recent emission trends? Please clarify.
- The statistics were given on Fig. 5 (see below) which showed that simulated SO$_2$ in Beijing is overestimated by nearly 300%. Even after considering the SO$_2$ reduction by heterogeneous reactions, the SO$_2$ emissions in Beijing would need to

be reduced by 60-70% to match the observations. We think the overestimation can be explained by the recent emission trends. As shown in Krotkov et al. (2016), around 30% $SO_2$ vertical column densities reductions were observed from OMI over Eastern China for 2010-2014, and the reduction ratio reaches 50-60% for the period 2008-2014. As the MEIC-2010 emissions inventory relied on the annual statistical books in which the data is often 2-3 years older than the actual year. We assumed that the $SO_2$ emission levels in MEIC-2010 were closer to the previous 2-3 years (2007-2008). The reductions in Beijing are likely larger than the Eastern China average since more strict measurements were implemented in Beijing. We have added the discussion in the text (Page 11, lines 10-18).

[Figure]

8) Page 11, second paragraph: Some discussions on the use of observed $SO_2$ concentrations to calculate model SOR are needed. Despite the sulfur rich environment, reducing $SO_2$ emissions in the model not only reduces $SO_2$ concentrations, but also aerosol sulfate concentrations. How would SOR respond to $SO_2$ emission changes in the

model? This can be evaluated with the simulation with 25% $SO_2$ emission reduction. I suggest add some sentences discussing the uncertainties in the model SOR values.

- Thanks! We have added the discussion in the text (Page 12, lines 32-35).

9) Page 25, Line 25 "We conclude that RH in the 80-100% range is a significant factor contributing to peak $PM_{2.5}$ values". The conclusion is only partly true. In the 80-100% range SOR and NOR values are much higher, but as for the peak $PM_{2.5}$ values, from Figure 8, it appears that there are comparable amounts of high $PM_{2.5}$ values in the 60-80% range. Please clarify.

- Thanks! We have changed the text.

Technical Comments:
- Thanks! We have made corrections/revisions below according to the suggestions.

1) Page 9, Line 2-4 "But correlations for boundary layer height and 10-m wind speed", missing some words here? What correlations?

2) Page 10, Line 9 "Since there" should be "Since their"?

3) Page 18, Figure 1 The blue symbol and the city labels are too small to read on the Figure. Please make them larger.

4) Page 20, Figure 3 Please describe in the Figure caption what are those meteorological variables, such as T2, RH2, WS10, and WD10.

[revised manuscript text omitted]

---

## Author Comment (AC2) · 21 Jul 2016

**Reviewer 1#**
The manuscript by Chen et al. considers factors that may help explain deficiencies in WRF-Chem simulations of secondary inorganic aerosol in extreme haze events in China. Following studies with other models, they consider additional heterogeneous reactions, and perform sensitivity studies to evaluate the impacts of these reactions as well as uncertainties in emissions. The topic is timely and of importance / relevance for ACP. The paper is generally well written, despite some grammatical issues. More quantitative comparison could be made to recent papers that have evaluated $SO_2$ and $NO_2$ trends in this region, or that have estimated the contribution of different aerosol precursor emissions to $PM_{2.5}$ in Beijing. The final model performance is indeed much better, although there is still room for improvement in the model and in our understanding of these haze events. I recommend publication following revisions to address the comments below.

We thank Referee # 1 for their comments and suggestions that have helped to improve this manuscript. Our responses to comments and the corresponding changes to the manuscript are detailed below in blue text. Revised manuscript is after the response letter.

Comments:
General: This work seems to still be missing a key reaction, which is aqueous-phase oxidation of S(IV) (the sum of dissolved $SO_2$, $HSO_3$ and SO32 ) by dissolved nitrogen dioxide ($NO_2$) that has been documented in the literature (Lee and Schwartz 1983, Clifton et al., 1988, Sarwar et al., 2013). As shown in Zhang 2015b, this made a substantial improvement to GEOS-Chem (in ways which would likely similarly improve the WRF-Chem) beyond the heterogeneous reactions that are considered here. Thus, I would also suggest the authors include this reaction in their analysis as an additional sensitivity calculations.

- Thanks for the suggestion! We have added the relevant aqueous reactions in WRF-Chem model following the study of Zhang et al. (2015b) and did another sensitivity simulation AQ_BASE. As there is not much precipitation and clouds during our simulation period, only the reactions ($H_2O_2$, $O_3$ and $NO_2$) on deliquescent aerosols are added in AQ_BASE. The main parameters, including the pH values are all the same as in Zhang et al. (2015b).
- The figure below shows the spatial distribution of mean sulfate (in $PM_{2.5}$) for the period of October 15-31, 2014 over the NCP (upper panels) and over the whole domain (lower panels). Those aqueous reactions do increase the sulfate concentrations by 13-15% even for the high sulfate region in central China and Sichuan. For Beijing, the newly added aqueous reactions improve the sulfate concentrations by 4.9% during clean days and 9.9% during heavy polluted days at the BNU site.
- In the study of Zhang et al. (2015b), they show 30% increases over the NCP by adding those reactions on deliquescent aerosols (confirmed by personal communication with the author). Our results are similar to their study, but the sulfate increase is a little bit lower. These differences likely depend on how well

WRF-Chem estimates aerosol water content as well as differences in RH between the two studies.
- We have added the discussions in the text (Page 13, third paragraph)

[Figure]

Figure 10. Spatial distribution of mean sulfate (in PM$_{2.5}$) from HET_BASE and AQ_BASE (units: μg m$^{-3}$) for the period of October 15-31, 2014.

3.5: There are several recent papers on $SO_2$ and $NO_2$ trends, for example Krotkov et al., ACP, 2015, see Fig 8, or Cui et al., ACP, 2016. The former would be useful to compare to when considering the $SO_2$ and $NO_2$ emissions trends projected in this paper.

- We have added the discussion in the text (Page 3, first paragraph)

3.11: GEOS-Chem was also used to specifically quantified the role of NH3 in Zhang et al. 2015b.

- Thanks. We have added the discussion in the text.

3.6: Not clear what is meant by "published paper". Perhaps official report? Or bottom up inventory?

- We originally referred to the reports or published papers of bottom-up emission inventories for recent years. But we have corrected the text as we did find one reference (Xia et al., 2016) that reported bottom-up emission rate changes in recent years (Page 3, lines 6-14).

p4/Table 1: What scheme is used for calculating gas-aerosol partitioning of HNO3/NO3 and NH3/NH4?

- The MOSAIC aerosol model (Zaveri et al., 2008) performs those calculations.

5.8: One would reach the same conclusion in this particular case, but more rigorously the moles of NH3 should be compared to the moles of 2 x $SO_2$ + $NO_x$.

- Thanks. We have given the emissions in moles in the text and also revised the text (Page 5, lines 6-10).

5.13: Recent Nature Geo paper (McLinden et al., 2016) highlights missing $SO_2$ sources in this (or similar) inventory.

- Yes. There might be some missing $SO_2$ sources in the released MEIC EI or similar EIs especially in XinJiang and Inner Mongolia since new power plants were installed recently (although detailed point source information is not updated). Since our study focuses on the NCP, those missing sources would not likely affect our results. We have emphasized in the text that the conclusions are based on the NCP region only.

8.20: Could some comparison to other studies / domains / models be referenced here, in terms of substantiating what it to be considered a "reasonable" accuracy for this type of model? At present, that word is used rather loosely.

- Yes, we added a discussion comparing our results to Wang et al. (2016) and emphasized that the overestimated wind speed and underestimated RH may lead to biases in the chemistry simulations (Page 9, lines 2-13).

10.15: Well, that would depend on the NOx/VOC regime, which the authors could easily check from their modeling results.

- Thanks! The 34 monitoring sites include some in Beijing urban area (VOC-limited regime) and also some in the suburban region ($NO_x$-limited regime). The averages of them show a back-and-forth shift between the two photochemical regimes of $O_3$ production. Thus, the comparisons to the averages of those sites cannot be used as evidence to make conclusions regarding the $NO_x$ emissions. We have corrected the text (Page 11, lines 5-10).

10.17: This could also instead indicate that $SO_2$ oxidation is too weak / slow in the model.
- Thanks! We have added the discussion in the text.

Section 4: It wasn't clear to me why the detailed speciated analysis was limited to only a few days. Why was this not performed for the entire month? Were the observations just not available? The peak $PM_{2.5}$ concentrations earlier in the month, Oct 7 - 10, were the largest of the month, and at a time when RH was well simulated in the model. Seems like this would be a good target to include in the analysis.
- The observations were only available after Oct. 15 since the campaign targeted the APEC period (in early November). No observations were made in early October.

11.24: The average magnitude is improved, the temporal correlation is not likely improved. Can the authors provide a table, or perhaps just write directly on these plots, what the statistics such as R^2 and NMB are for these results?
- Thanks! We have added the statistics in Fig. 5 and 6.

Fig 7: In terms of comparing the observed to modeled % contributions from sulfate/ammonium/nitrate, it would easier to evaluate visually if the plots were of just these 3 species. At the very least, they could remove CL from the obs, so make a more direct comparison.
- Thanks! We have removed CL from the obs in Fig. 7.

12.27: This could be understood more quantitatively by considering results from Zhang 2015b.
- Thanks! We have cited the paper in the discussion.

General: Did the authors try increasing the RH as a sensitivity test?
- We did not perform a test to increase RH. Instead, we used fixed high reaction coefficients for sensitivity testing purposes. As for the underpredictions in RH, data assimilation is likely the best approach to improve the simulation. This approach may be used in future research.

Corrections: General: I didn't type up all of the grammatical corrections; please have Jerome do a final proof-read of the article prior to resubmission.

- Thanks! We have corrected those grammatical errors listed below and also did thorough proof-read.

abstract, last line: situations -> concentrations

2.4: exceeding the WHO standard tenfold

2.15: PM2.5, the formation

3.1: 2014 may not be reflected (or are not reflected)

3.15: WRF-Chem and

3.16: conducted simulations  To our best knowledge

3.17:WRF/Chem model.  -> WRF-Chem.

3.18: using available

3.22: analysis for

3.28:missing comma

3.35: et al.,

4.19: nonvolatile, the

5.14: from two other aspects

5.36: fall into

6.8: respectively,

6.17: include equation number, comma goes directly after the equation on the same line, and then "Where" is not capitalized.

6.21: units of surface area per unit volume of air seem incorrect.

7.8: we first

7.10: simulations; we then tested::

[revised manuscript text omitted]
 $2\times10^{-5}/5\times10^{-5}$; SO2_only_2 with lower/upper limits $1.0\times10^{-4}/2.6\times10^{-4}$.